# Structure and energy transfer of a far-red–absorbing euglenophyte PSI–LhcE–LhcbM supercomplex

Kang Li[1,2,3,6], Bing-Yue Qin[1,6], Yu-Zhong Zhang [1,2,3,6], Hao-Jie Wang[1], Quan Wen[4], Xin-Xiao Qu[1], Fang Zhao[1,2], Xiu-Lan Chen [1,3], Jun Gao [4], Lu-Ning Liu [2,5] & Long-Sheng Zhao [1,3] ✉

Euglenophyta originated from a secondary endosymbiosis between a phago-trophic euglenid and a green alga. Euglenophytes acquired photosynthesis-related genes from diverse algal lineages, representing a remarkable example of plastid evolution in the green lineage. Here, we solve the structure of the PSI–LhcE–LhcbM supercomplex from the euglenophyte *Euglena gracilis*. This supercomplex contains a simplified PSI core and an extensive antenna system, including 13 LhcEs and 2 LhcbMs. The LHCs are arranged as centrosymmetric dimers or monomers, resulting in a specific antenna organization. Notably, the LhcbMs are robustly integrated into the supercomplex through direct inter-actions with PsaB, PsaJ, and PsaF, without the need for phosphorylation. This phosphorylation-independent assembly mechanism highlights a specific adaptation in euglenophyte PSI–LhcE–LhcbM organization. We also identify specific structural features surrounding red-shifted chlorophyll *a* pairs in LHCs, which may account for the enhancement of far-red light absorption of PSI–LhcE–LhcbM. Computational simulations further reveal a distinctive pig-ment network, facilitating efficient energy transfer within the supercomplex. Our study not only provides insights into the mechanisms of light harvesting and energy transfer in euglenophyte PSI–LhcE–LhcbM but also broadens the framework of plastid evolution and complexity, with implications for mod-ulation and bioengineering of photosynthetic complexes.

Oxygenic photosynthetic organisms, including cyanobacteria, algae, and land plants, are fundamental to life on Earth, supplying essential substances and energy through photosynthesis while maintaining ecological balance[1]. Photosynthetic light energy conversion is syner-gistically driven by protein complexes in thylakoid membranes. Among them, photosystem I (PSI) and photosystem II (PSII), as multisubunit pigment-protein complexes, perform the functions of light capture, excitation energy transfer and electron transfer[2,3]. In photosynthetic eukaryotes, PSI associates with membrane-embedded light-harvesting complexes (LHCs), forming PSI–LHC supercomplex. To date, various structures of the PSI–LHC supercomplex have been reported[4–29]. The PSI core is relatively conserved and exists as a

[1]Marine Biotechnology Research Center, State Key Laboratory of Microbial Technology, Shandong University, Qingdao, China. [2]MOE Key Laboratory of Evolution and Marine Biodiversity, Frontiers Science Center for Deep Ocean Multispheres and Earth System & College of Marine Life Sciences, Ocean University of China, Qingdao, China. [3]Laboratory for Marine Biology and Biotechnology, Qingdao Marine Science and Technology Center & Laoshan Laboratory, Qingdao, China. [4]Hubei Key Laboratory of Agricultural Bioinformatics, College of Informatics, Huazhong Agricultural University, Wuhan, China. [5]Institute of Systems, Molecular and Integrative Biology, University of Liverpool, Liverpool, UK. [6]These authors contributed equally: Kang Li, Bing-Yue Qin, Yu-Zhong Zhang. ✉e-mail: zhaols@sdu.edu.cn

monomer with occasional additions and the loss of specific subunits. However, the size and arrangement of LHC systems vary among species, reflecting the adaptation of PSI–LHC to environmental changes and evolutionary pressures to regulate light energy conversion[28–30].

Euglenophyta, a group of predominantly unicellular organisms that lack cell walls, exhibit a blend of plant and animal characteristics. Its eyespot can sense light and help it quickly avoid unfavorable lighting conditions. Plastid-containing species perform autotrophic photosynthesis but can also engage in heterotrophic nutrition via phagocytosis or pinocytosis[31]. Euglenophytes possess a substantial array of genes, particularly those associated with photosynthesis and plastid functions, which appear to have been acquired from diverse algal lineages[31–34]. Their plastids, enveloped by three membranes, trace their origins to a secondary endosymbiotic event with a green alga closely related to extant members of the *Pyramimonas* genus, approximately 500 million years ago[31,35]. Euglenophytes also hold genes of red-lineage origin acquired by horizontal gene transfer or a past endosymbiosis with red-lineage algae, and some of these genes seem to facilitate the successful integration of green-algal endosymbionts[32]. The presence of pigments like diadinoxanthin, which is absent in the green lineage but found in the red lineage, further suggests a complex evolutionary trajectory of euglenophytes[36]. Mixotrophic nutrition, rapid light response, and chimeric genome may lead euglenophyte photosystems to feature specialized structural adaptations for light capture and energy transfer.

Recently, a chlorophyll (Chl) *a* far-red–absorbing antenna complex, composed of a species-specific LhcE protein family, was identified in *Euglena gracilis*, a model organism for photosynthesis studies of Euglenophyta[37]. Intriguingly, the antenna system of *E. gracilis* PSI consists of LhcE proteins and LhcbM proteins, differing from other green-lineage PSI–LHCI.

Here, we report the cryo-electron microscopy (cryo-EM) structure of the PSI–LhcE–LhcbM supercomplex from *E. gracilis*. The supercomplex consists of a simplified core with 8 subunits and 15 peripheral LHCs, including 12 LhcEs and 3 LhcbMs, which together assemble into a specific and stable PSI–LhcE–LhcbM supercomplex. Notably, the LhcbMs lack phosphorylated residues and form strong interactions with the PSI core, enabling their robust assembly with PSI independent of phosphorylation. Furthermore, unlike other characterized green-lineage LHCs, euglenophyte LHCs form centrosymmetric dimers or monomers, and utilize diadinoxanthin as their major carotenoid. These protein and pigment organizational features enhance far-red light absorption of the PSI–LhcE–LhcbM supercomplex, lay the foundation for specific excitation energy transfer pathways within the PSI–LhcE–LhcbM and highlight the specific evolutionary position of euglenophyte PSI–LHC during endosymbiosis.

## Results and discussion
### Overall structure

The PSI–LhcE–LhcbM supercomplex was isolated from *E. gracilis* and was characterized by absorption spectroscopy, electrophoresis, and pigment analysis using high-performance liquid chromatography (HPLC) (Supplementary Fig. 1). The PSI–LhcE–LhcbM structure, with a molecular mass of ~761 KDa, was solved at an overall resolution of 2.72 Å using single-particle cryo-EM analysis (Supplementary Fig. 2 and Supplementary Table 1). The monomeric PSI core comprises 8 subunits, including PsaA-F, PsaJ and PsaM (Fig. 1), in line with recent phylogenomic and proteomic analysis[37]. Surrounding the PSI core is a

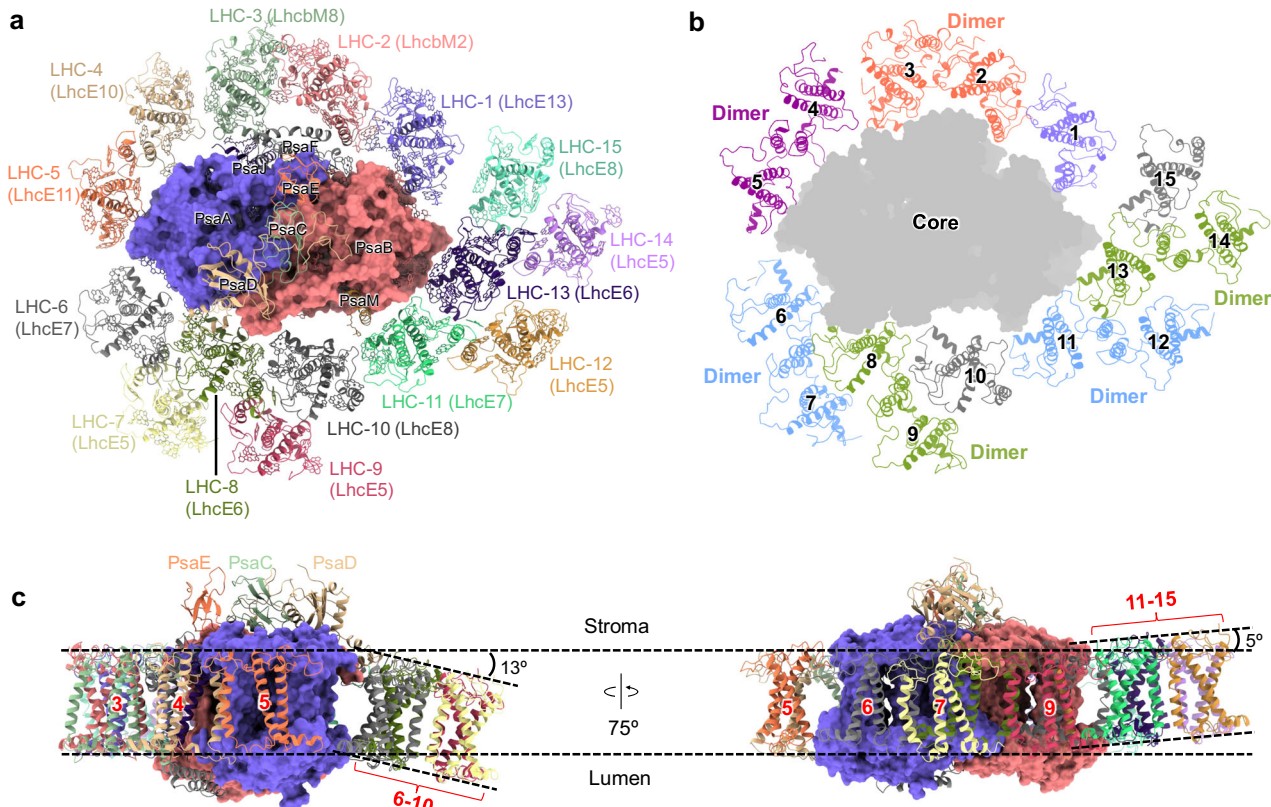

**Fig. 1 | Overall architecture of the euglenophyte PSI–LhcE–LhcbM supercomplex. a** PSI–LhcE–LhcbM supercomplex viewed from stromal side. Core subunits are labeled as their names. LHCs are labeled as LHC-1 to LHC-15. The protein family of each LHC is indicated in bracket. **b** Six LHC dimers within the euglenophyte PSI–LhcE–LhcbM supercomplex. Numbers indicate the 15 LHCs around the PSI core. **c** Side view of the PSI–LhcE–LhcbM supercomplex. Numbers indicate the LHCs around the PSI core. The tilt angles of LHC-(6-10) and LHC-(11-15) relative to the PSI core are indicated.

large antenna system of 15 LHCs, which is larger than the antenna systems of any green-lineage PSI–LHCIs (4-10 LHCIs) characterized to date and is comparable in size to the antenna systems of *Ostreococcus tauri* PSI–LHCI–Lhcp (15 LHCs) and *Chlamydomonas reinhardtii* PSI–LHCI–LHCII (16 LHCs)[4–14]. Our structure shares similar features with the recently reported structure of *E. gracilis* PSI–LHC, whereas two more LHCs (LHC-14/15) were identified in our structure[38]. Notably, the majority of euglenophyte LHCs are arranged as dimers, including five LhcE dimers and one LhcbM dimer, which are predominantly positioned on the PsaD-PsaM side (Fig. 1b). This results in a specific antenna organization that is distinct from those found in other green-lineage PSI–LHCs. The association of specific LhcE proteins contribute to the far-red absorption of the PSI–LhcE–LhcbM supercomplex, leading to a red absorption shoulder relative to PSII–LHC super-complex and a predominant 77 K fluorescence emission peak at 732 nm[37,39]. We dissociated LHCs from the PSI–LHC by treating the PSI–LHC sample with 1.5% β-DDM. Through sucrose density gradient ultracentrifugation, two major bands were identified (Supplementary Fig. 1a). The absorption spectra show that Band 2 has a lower absorption than the untreated PSI–LHC at 400-500 nm, indicating that Band 2 contains lower amount of Chl *b* and carotenoids (Supplementary Fig. 1b). SDS-PAGE shows that Band 1 is mainly composed of LHC proteins and Band 2 contains less LHCs compared with untreated PSI–LHC (Supplementary Fig. 1c, d), indicating that part of the LHCs were dissociated from PSI–LHC. The differential absorption spectrum between untreated PSI–LHC and Band 2 exhibits a far-red shoulder, demonstrating that LHCs associated with the PSI core could enhance far-red light absorption, in line with reported results. Moreover, the membrane-spanning region of euglenophyte PSI–LhcE–LhcbM exhibits curvature, with LHCI-(6-10) and LHCI-(11-15) curving in opposite directions: LHCI-(6-10) pivots towards the lumenal side, whereas LHCI-(11-15) pivots towards the stromal side, relative to the core (Fig. 1c). Additionally, 244 Chl *a*, 27 Chl *b*, 37 diadinoxanthin (Ddx), 1 neoxanthin (Neo), 13 β-carotene (β-Car), 2 naphthoquinones, 3 $Fe_4S_4$ clusters, and 16 lipid molecules were identified in the structure (Supplementary Fig. 3 and Supplementary Table 2). The ratio of assigned pigments in our structure (Chl *a*: Chl *b*: Diadinoxanthin: Neoxanthin = 100: 11: 15: 0.4) is roughly comparable with that biochemically determined in *E. gracilis* PSI–LHC (100: 9.4: 26: 1.3)[37].

## Structure of the euglenophyte PSI core

The euglenophyte PSI core lacks six peripheral subunits (PsaG, PsaH, PsaI, PsaK, PsaL, PsaN, and PsaO), which are typically present in green algae (Supplementary Fig. 4 and Supplementary Table 3). Genomic analysis confirms the absence of the corresponding genes in *E. gracilis*[34]. This simplified PSI configuration is reminiscent of the "mini-PSI" observed in the salt-tolerant alga *Dunaliella salina* under high-salt conditions, where only seven core subunits are retained[7]. However, *D. salina* also possesses a large-PSI form containing the full complement of peripheral subunits[8]. Thus, the euglenophyte PSI core represents the smallest known PSI core at the gene level.

Intriguingly, an additional domain containing three short helices was identified at the N-terminus of PsaD (Fig. 2a). This structural domain form interactions with LHC-6 and LHC-8 (Fig. 2b), indicating that euglenophyte PsaD not only facilitates the docking of ferredoxin, as is typical, but also plays a direct role in anchoring peripheral LHC complexes, effectively compensating for the absence of several peripheral PSI subunits. *E. gracilis* PsaD is likely derived from cyanobacteria, in contrast to other PSI core subunits that are originated from green algae[38]. The PSI core possesses the PsaM subunit, which is also present in the PSI cores of green algae *Bryopsis corticulans* and *O. tauri*[4,9] (Supplementary Fig. 4e) but is absent in *D. salina* and *C. reinhardtii*[5–8]. Euglenophyte PsaM is associated with a Car pigment, similar to *O. tauri* PsaM but not for *B. corticulans* PsaM. Moreover, a swing occurred on the lumenal head group of this Car compared with

that of *O. tauri* PsaM, because a Chl *a* molecule, which was observed in *O. tauri* LHCI, is absent in euglenophyte PSI–LhcE–LhcbM (Supplementary Fig. 4e).

Most euglenophyte PSI core subunits retain conserved pigment-binding sites except for PsaJ, which harbors an additional Car802 not found in green algal PSI cores (Supplementary Fig. 5). This Car molecule is located between the Chls at the interface of the PSI core and LHCII-3, suggesting its potential role in mediating energy transfer and dissipation (Supplementary Fig. 5i). Additionally, several β-Cars in the PSI core, including PsaA 835/836, PsaF 805, and PsaJ 801/802, are substituted by Ddx, a feature reminiscent of the diatom PSI core (Supplementary Fig. 5b)[25,26].

## Structures of euglenophyte LHCs

Euglenophyte LHCs possess three transmembrane (TM) helices (αA, αB, and αC) and three hydrophilic helices (αD, αE, and αF) on the lumenal surface (Fig. 2c), resembling the structures of green algal LHCs[4–9]. The αA and αB helices are highly conserved, whereas αC varies in its tilt direction (Fig. 2c, Supplementary Fig. 6, and Supplementary Fig. 7). Most terminal loops and inter-helix loops share similar structures, except for the loops of LHC-1/2/3, the AC loops (between αA and αC) of LHC-5/10/15, and the CE loop (between αC and αE) of LHC-4 (Fig. 2c).

A recent study classified *E. gracilis* LHC proteins into over 20 distinct Lhc families, including LhcsRL, Lhcb4, Lhcb7, LhcbM1-3, LhcbM5-8, LhcbMX1-3, LhcbMX5, and LhcE1-13, through phylogenetic analysis of 158 predicted LHC proteins from *E. gracilis* and a broader reference set of Viridiplantae genomes[37]. The LHC proteins of LhcE5/6/7/8/10/11/13 families and LhcbM2/M8 families were assigned in our *E. gracilis* PSI–LHC structure based on the best match between the amino acid sequence and the cryo-EM map, in line with recent mass spectrometry analysis[37]. The gene names corresponding to these LHCs were not assigned in recently reported *E. gracilis* PSI–LHC, as the comprehensive investigation of the phylogenetic distribution of *E. gracilis* LHCs had not been available before their publication[38]. Phylogenetic analysis revealed that LHCs in *E. gracilis* PSI–LHC are more closely related to green algal Lhcbs rather than to green algal Lhca and red-lineage LHCs (Supplementary Figs. 8 and 9)[37,38,40,41]. LhcbM-type LHC-2/3 and LhcE-type LHC-1 possess similar structural features to green algal LhcbM. Specifically, LHC-2/3 share nearly identical structures with green algal LhcbM (Supplementary Fig. 10a). While LhcE-type LHC-4 to LHC-15 also exhibit considerable structural similarities with green algal LhcbM, they display some variations in the loop structures (Supplementary Fig. 10b). LHC-1 and LHC-4 to LHC-15 possess identical N-terminal structures, which are shorter than that of LhcbM (Fig. 2c and Supplementary Fig. 10b). Notably, the CE loop of LHC-4 is substantially longer than those of other LhcE-type LHCs (Supplementary Fig. 10b).

Two distinct dimeric conformations of euglenophyte LHCs are identified: The LhcbM dimer (LHC-2/3) exhibits the same assembly patterns with LhcbMs in green algal LHCII trimers, whereas other five LhcE dimers (LHC-4/5, LHC-6/7, LHC-8/9, LHC-11/12 and LHC-13/14) share similar assembly patterns with each other, with a central rotational symmetry (Fig. 2d, e), which differs from the asymmetrical LHCI dimer identified at the PsaI-PsaM side in green algae[4–6,8,9]. These symmetric dimers are stabilized by interactions between the αC helices and the AC loops of LhcE monomers (Fig. 2e). The αC helices are oriented more perpendicularly to the membrane than those in LhcbM, and the AC loops exhibit distinct structural features from those of monomeric LhcEs (Supplementary Fig. 10c, d). These structural variations facilitate the formation of symmetric LhcE dimers. In these dimers, the AC loop of each LhcE extends toward the αC helix of the opposing LhcE, facilitating interactions between adjacent LhcEs (Fig. 2e). Interestingly, while symmetric LHC dimers were also found in diatom and haptophyte PSII–LHCII, the distance

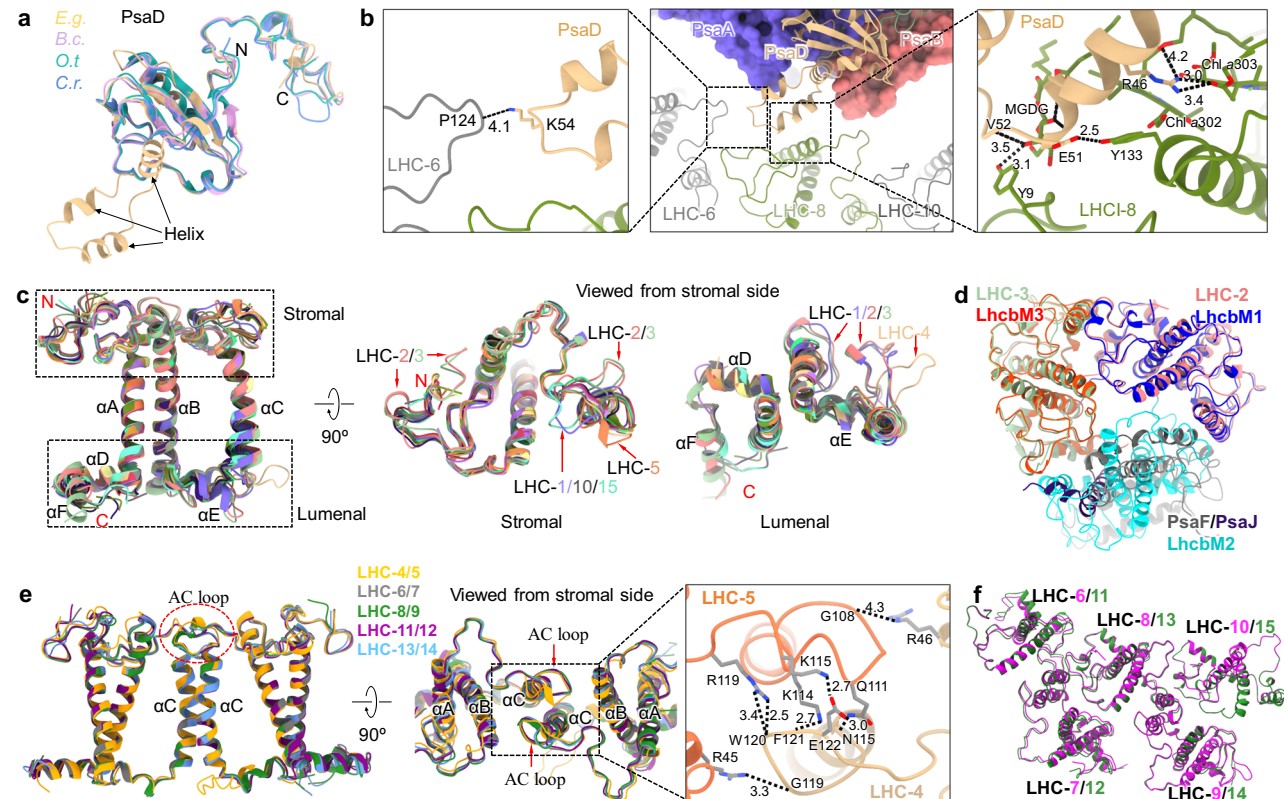

**Fig. 2 | Structural features of the PSI core subunits and euglenophyte LHCs.**
**a** Comparison of PsaD of euglenophyte *Euglena gracilis* (*E.g.*) with green algae *Bryopsis corticulans* (*B.c.*), *Ostreococcus tauri* (*O.t.*), and *Chlamydomonas reinhardtii* (*C.r.*). **b** The locations of PsaD in PSI core viewed from the stromal side and its interactions with adjacent LHCs. Squared areas are enlarged, showing the detailed hydrogen bond interactions. Interactions are indicated by black dashed lines with distances labeled in Å. **c** Structural superposition of 15 euglenophyte LHCs. The N-terminus, C-terminus, and three transmembrane helices are labeled as N, C, αA, αB,

and αC. The helices at lumenal surface are labeled as αD, αE, and αF. The squared areas are enlarged. Specific loop structures are indicated. **d** Structural comparison of LhcbM-type LHC-2/3 dimer–PsaJ/F with green-algal LhcbM trimer. **e** Comparison of the symmetric dimers. Squared areas are enlarged, showing the detailed hydrogen bond interactions. Interactions are indicated by black dashed lines with distances labeled in Å. **f** Comparison of the two identical structural modules: module 6–10 and module 11–15.

between LhcEs in euglenophyte LhcE dimers is notably larger (Supplementary Fig. 11)[42,43]. In addition, diatom and haptophyte LHC dimers have relative distant genetic relationships with euglenophyte LHC dimers (Supplementary Fig. 9).

## Organization of LHCs and assembly of LhcEs in the PSI–LhcE–LhcbM supercomplex

The antennae of euglenophyte PSI–LhcE–LhcbM comprise three LHC monomers and six LHC dimers, exhibiting a highly distinctive arrangement compared to green algal PSI–LHCs (Fig. 1a, b and Supplementary Fig. 12a–f). Eleven LHCs, including two LhcbMs and nine LhcEs, associate directly with the PSI core, forming a closed ring. On the PsaA-PsaJ-PsaF-PsaB side, one LhcE monomer (LHC-1), one LhcbM dimer (LHC-2/3), and one LhcE dimer (LHCI-4/5) are arranged in a single layer (Fig. 1b). The specific assembly patterns of LHC-1/2/3/4/5 lead to one more LHC associated directly with the PSI core at the PsaA-PsaJ-PsaF-PsaB side compared to green-lineage PSI–LHCs (Supplementary Fig. 12). On the PsaA-PsaD-PsaM-PsaB side, two LhcE monomers (LHC-11 and LHC-15) and four LhcE dimers (LHC-6/7, LHC-8/9, LHC-11/12 and LHC-13/14) form a double-layered arrangement. This specific antenna arrangement dramatically differs from those of green-lineage PSI–LHCI–LHCIIs (Supplementary Fig. 12a–c)[4–9]. Intriguingly, LHC-6/7/8/9/10 and LHC-11/12/13/14/15 form two identical structural modules: Module 6-10 and Module 11-15 (Fig. 2f). This distinctive modular structure further distinguishes the euglenophyte PSI antenna system from its green algal counterparts. In addition, the LHC

arrangements in PSI–LhcE–LhcbM are distinct from those found in red-lineage PSI–LHCIs (Supplementary Fig. 12g)[21,23,25,27], highlighting their evolutionary divergence.

The assembly of euglenophyte LhcEs shows remarkable variations compared to the regular arrangement of green algal Lhcas (Supplementary Fig. 12). The LHC-4/5 dimer interacts with the PSI core without the assistance of PsaK. LHC-5 displays a counterclockwise rotation relative to Lhca3 in green algal PSI–LHCI (Supplementary Fig. 12a). The specific terminal loops of LHC-5 and the long CE loop of LHC-4 mediate the binding of LHC-4/5 dimer with the PSI core, and additional stabilization is provided by contacts between the AC loop of LHC-4 and the N-terminal loop of LHC-3 (Supplementary Fig. 13a).

The binding position of the LHC-6/7/8/9/10 module to the PSI core is similar to that of the Lhcp trimers in *O. tauri* PSI-LHCI-Lhcp supercomplex and LHCII trimers of *C. reinhardtii* PSI-LHCI-LHCII (Supplementary Fig. 12a, f)[9,13,14]. However, LHC-6/7/8/9/10 directly associate with the PSI core without the mediation of PsaK/O/L/H, whereas PsaK/O/L/H in *O. tauri* and *C. reinhardtii* are crucial for the binding of Lhcp trimers and LHCII trimers, respectively[9,13,14]. The LHC-11/12/13/14/15 module occupies the positions analogous to those of LHCI dimers in green algal PSI–LHCI (Supplementary Fig. 12d–f)[4–9]. The four dimers (LHC-6/7, LHC-8/9, LHC-11/12, and LHC-13/14) share the same orientations. In these dimers, LHC-6/8/11/13 directly associate with the PSI core through interactions between their N-termini/AC loops/C-termini/BE loops and the surface loops of PsaA/B/D/M (Supplementary Fig. 13b, c), distinct from the binding pattern of the LHC-4/

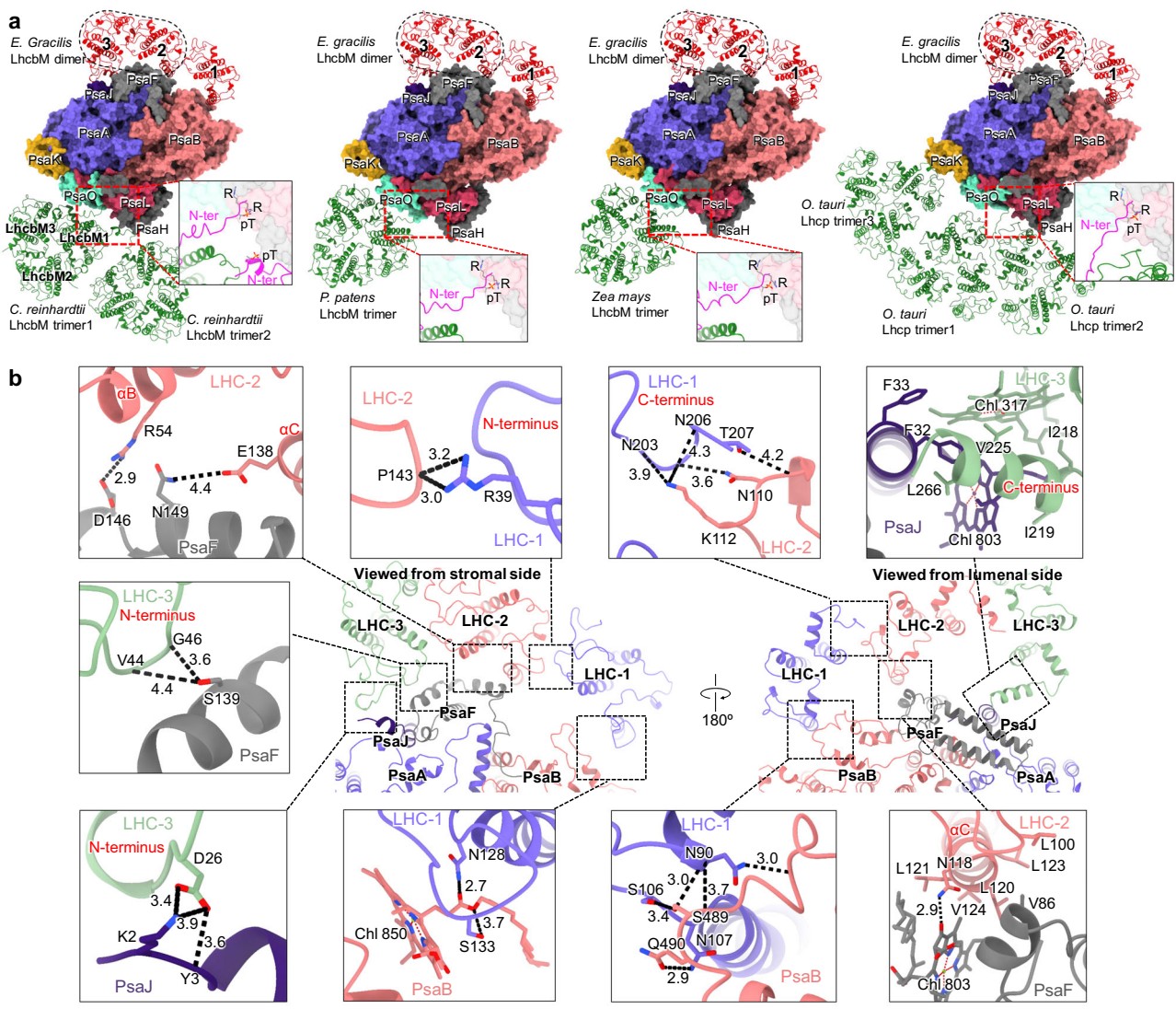

**Fig. 3 | Locations of euglenophyte LhcbMs and their interactions with the PSI core, and comparison with those of LhcbMs in other green-lineage PSI–LHCI–LHCII and Lhcps in PSI–LHCI–Lhcp from green algae *Ostreococcus tauri*. a** The locations LhcbM dimer in PSI–LhcE–LhcbM from euglenophyte *Euglena gracilis* and comparison with those of LhcbM trimers in PSI–LHCI–LHCII from green alga *Chlamydomonas reinhardtii* (PDB: 7DZ7), moss *Physcomitrium patens* (PDB: 7XQP), and land plant *Zea mays* (PDB: 5ZJI), and Lhcp trimers in PSI-LHCI-Lhcp from green algae *Ostreococcus tauri* (PDB: 7YCA). The enlarged view shows the elongated N-terminus of Lhcb in LhcbM trimers and Lhcp in Lhcp trimers and RRpT (pT, phosphorylated Thr residue) motif in the N-terminus, associating with the PSI core through interactions with PsaO/L/H. **b** Interactions between LHC-1/2/3 and the PSI core and between LHC-1 and LHC-2/3 dimer. Squared areas are enlarged, showing the detailed hydrogen bond interactions and hydrophobic interactions. Interactions are indicated by black dashed lines with distances labeled in Å.

5 dimer. LHC-6/8/10/11/13/15 possess distinct orientations, facing towards the PSI core at the opposite sides, compared to those typically observed in green algal LHCI (Supplementary Fig. 12). The close interactions among LHCs in the structural module facilitate their binding with the PSI core. Interactions between LHC-6/7 dimer and LHC-8, LHC-8/9 dimer and LHC-10, LHC-10 and LHC-11, LHC-11/12 dimer and LHC-13, and LHC-13/14 dimer and LHC-15 possess similar patterns (Supplementary Fig. 13d). For instance, the AC loop of LHC-6 and the N-terminal loop of LHC-7 form interactions with the N-terminal loops of LHC-8 (Supplementary Fig. 13e). In addition, the extended C-terminal loops of LHC-8/10/13/15 enable interactions with adjacent LHCs (Supplementary Fig. 13e).

### Assembly of LhcbMs in the PSI–LhcE–LhcbM supercomplex
It is generally believed that the LHCII antennae (LhcbMs) associated with PSI–LHCI–LHCII complexes are involved in state transitions in green algae and land plants, which balance the excitation energy

between PSI and PSII[13–15]. In state 1, LHCIIs are predominantly associated with PSII, channeling the excitation energy to the PSII core. In state 2, a subset of LHCIIs undergo phosphorylation and subsequently migrate to PSI, resulting in the formation of the PSI–LHCI–LHCII supercomplex to facilitate energy transfer to PSI and prevent photodamage to PSII. However, LHCIIs have been identified as constitutive antennae in the large PSI supercomplex from low-light-adapted moss, which was not found to be involved in state transitions[16,17,44,45]. In addition, Lhcp antennae were determined to associate with *O. tauri* PSI–LHCI under low-light conditions, forming PSI–LHCI–Lhcp supercomplex[9]. Intriguingly, our structural analysis reveals that the assembly patterns, binding positions and association domains of euglenophyte LhcbMs are distinct from those of LhcbMs in other PSI–LHCI–LHCII supercomplexes and Lhcps in PSI–LHCI–Lhcp supercomplex characterized (Fig. 3a)[13–17].

The LhcbMs in euglenophyte PSI–LhcE–LhcbM are present as a dimer positioned at the PsaJ-PsaF side, whereas LhcbMs in other

characterized PSI–LHCI–LHCII supercomplexes and Lhcps in PSI–LHCI–Lhcp supercomplex bind to the PsaK-PsaO-PsaL-PsaH side as trimers (Fig. 3a). LhcbM trimers and Lhcp trimers associate with the PSI core through the elongated N-terminus of LhcbM and Lhcp, respectively, and the RRpT (pT, phosphorylated threonine residue) motif in the N-terminus, which form interactions with PsaO/L/H (Fig. 3a)[13–17]. In contrast, euglenophyte PSI–LhcE–LhcbM lacks the PsaO/L/H subunits and any elongated N-terminal regions, RRpT motif, and phosphorylated Thr residues within the LhcbMs (Fig. 1, Fig. 2c, and Supplementary Fig. 14). Instead, LhcbMs form strong interactions with the PsaB/J/F subunits and LHC-1 (Fig. 3b). LHC-1 binds directly to the PSI core through interactions with PsaB without the mediation of PsaG (Fig. 3b), resulting in a clockwise rotation of LHC-1 relative to Lhca1$_{IN}$ in green algal PSI–LHCI (Supplementary Fig. 12a)[4–9]. In addition, the terminal loops of LHC-1 stabilize its binding via interactions with neighboring LHC-2 (Fig. 3b). The PsaJ and PsaF subunits associate with the LhcbM dimer (LHC-2/3), forming an organization similar to that of green algal LHCII trimer (Figs. 2d and 3b). In oat seedlings, PsaF has been found to promote the association of LHCI and the functionality of PSI complexes, such as photoprotection[46]. The binding sites of the LhcbM dimer with the euglenophyte PSI core are opposite to those of LhcbM1/M3, suggesting their entirely distinct binding strategies (Fig. 3a). LHC-2 interacts with PsaF through its αB and αC helices, instead of the elongated N-terminus and phosphorylated Thr residue identified in LhcbM1 (Fig. 3b). LHC-3 forms strong interactions with both PsaJ and PsaF subunits via its terminal loops (Fig. 3b). Overall, these structural features indicate that the specific assembly of LhcbM with euglenophyte PSI results in the formation of a robust PSI–LhcE–LhcbM supercomplex. This assembly likely occurs through a phosphorylation-independent mechanism and is not related to state transitions. A canonical state transition mechanism involving LhcbM trimers is absent in *E. gracilis*. Instead, a state transition-like mechanism was identified[37]. A specific pentameric LhcE antenna complex, not the classical LhcbM trimer antenna, is dynamically associated with PSII and involved in this state transition-like mechanism. Thus, *E. gracilis* employs a distinct strategy for balancing excitation energy.

Overall, the euglenophyte PSI antennae exhibit a remarkable degree of structural diversity and uniqueness, highlighting evolutionary adaptations that distinguish them from the more conventional arrangements observed in green algal PSI systems.

## Pigment arrangement in euglenophyte LHCs

The LHCs in the euglenophyte PSI–LhcE–LhcbM supercomplex harbor 155 Chl *a* and 27 Chl *b* molecules, and each LHC binds 11–15 Chls (Supplementary Table 2). Twenty Chl-binding sites were identified in euglenophyte LHCs, fifteen (301–311, 313–316) of which are conserved across green algal LHCs and five (312, 317–320) are specific to euglenophyte LHCs (Fig. 4a)[4–9,40,41]. Chl *b* was tentatively identified in sites 301/304/307/310/311/313/315, consistent with that in green algal LHCs (Supplementary Fig. 15a). Except for Chl 316/317, the Chl-binding sites in LHC-1/2/3 (301-311, 313-315) are consistent with those in green algal LHCII trimers[40,41]. The Chl 316 site binds to the C-terminal loop of LHC-2 (Supplementary Fig. 16a). Similar site was also found in green algal LHCI[4–9]. The Chl 316 site is positioned between LHC-2 and LHC-3, potentially facilitating energy transfer within the LHC-2/3 dimer. The Chl 317 site, associated with the specific αF helix of LHC-3, is located at the interface between LHC-3 and the PSI core (Supplementary Fig. 16b). The Chl 315 site is absent in LHC-4 to LHC-15, as the CE loop for the binding of Chl 315 in LHC-1/2/3 is absent in LHC-4 to LHC-15 (Fig. 2c). The orientation of Chl 301 varies among euglenophyte LHCs. In LHC-2/3 dimer, Chl 301 shares a similar orientation with its counterparts in green algal LHCII trimers, whereas the orientation of Chl 301 in other LHCs is comparable in red-lineage LHCIs (Supplementary Fig. 16c).

The positions of the five additional Chl-binding sites vary among LHCs: Chl 312 in LHC-6/8/10/11/13/15, Chl 317 in LHC-3, Chl 318 in LHC-4, Chl 319 in LHC-5, and Chl 320 in LHC-6/8/11/13 (Supplementary Table 4). Chl 312 associates with αA of LHCs, resembling the Chl-binding site in red-lineage LHCs, albeit with distinct orientations (Supplementary Fig. 16d). Chl 318, located at the extended CE loop of LHC-4, sits between the LHC-4/5 dimer and the PSI core (Supplementary Fig. 16e), and aligns in a parallel configuration with Chl 814 and Chl 816 of PsaA, potentially enhancing energy transfer from the antenna to the PSI core (Fig. 4b). Chl 318$_{LHC-4}$, Chl 814$_{PsaA}$, and Chl 816$_{PsaA}$ form a Chl trimer, reminiscent of a Chl trimer in cyanobacterial PSI core which is contribute to the low-energy state of cyanobacterial PSI[47]. Thus, this Chl trimer may be low-energy Chls in euglenophyte PSI–LhcE–LhcbM. Chl 319 substitutes Chl 314, and binds to the C-terminus of LHC-5. Chl 319 is located in the large gap between LHC-5 and LHC-6, and is closer to LHC-6 than Chl 314 (Supplementary Fig. 16f). Chl 320 is associated with the αC helices of LHC dimers in modules 6-10 and 11-15 and sits between dimers and adjacent antennas (Supplementary Fig. 16g). All of these additional Chl-binding sites are positioned on the lumenal side and at the interface between LHCs and between LHCs and the PSI core, suggesting their potential roles in mediating energy transfer from antennas to the PSI core (Fig. 4b).

Euglenophyte LHCs contain 33 Cars, including 32 Ddx and 1 Neo; each LHC binds 2 to 3 Cars, fewer than the 4 Cars in LhcbMs of green algal LHCII trimers (Supplementary Table 4)[40,41]. Three Car-binding sites (401-403) were identified in euglenophyte LHCs and are conserved with those in green algal LHCs (Fig. 4c). LHC-1/2/3 possess 3 Cars, and the Car molecule close to the N-terminus and C-terminus which was found in LhcbMs is absent. Among the LHC-4 to LHC-15, only two Car-binding sites (401-402) were identified (Supplementary Fig. 15b). This reduction in Car content suggests a reduced capability of euglenophyte LHCs for blue/green light absorption and energy dissipation. In addition, the Ddx/Dtx cycle appears to be independent of non-photochemical quenching in euglenophytes[48], and these organisms rely on their eyespot for rapid phototactic responses to fluctuating light environments, potentially reducing their dependence on Car-mediated photoprotection[49]. Intriguingly, Ddx is the major Cars in euglenophyte LHCs, which is exclusive in red-lineage algae such as diatom and haptophyte and is absent in other green-lineage algae[4–12,23–27], suggesting that euglenophytes acquired Ddx through gene transfer from red-lineage algae during endosymbiosis.

## Structural features possibly enhancing far-red light absorption of euglenophyte PSI–LhcE–LhcbM

The specific LhcEs account for the far-red light absorption of euglenophyte PSI–LhcE–LhcbM[37]. The low-energy red Chl *a*305/*a*306 pairs in euglenophyte LhcEs, counterparts of the red Chl *a*609/*a*603 pairs in green algae and land plants, serve as the primary pigments for far-red light absorption. Multiple factors can affect the energy states and red-shifted absorption of Chl *a* pairs[50]. The surrounding environment can regulate the extent of spectral red shift in Chl *a*305/*a*306 pairs[50,51]. The specific centrosymmetric LhcE dimer structure positions the two Chl *a*305/*a*306 pairs in close proximity, forming a distinct arrangement pattern (Supplementary Fig. 15c). The amino acid residue environment surrounding the Chl *a*305/*a*306 pairs also differs significantly from that of the Chl *a*603/*a*609 pairs in green algae and plants (Supplementary Fig. 15c, d). The Chl *a*305/*a*306 pair in each monomer forms additional interactions with amino acids from the opposing monomer (Supplementary Fig. 15c). Consequently, a specific mechanism may have evolved to regulate the far-red light absorption of the Chl *a*305/*a*306 pair in euglenophyte LhcE dimers.

Recently, research has revealed structural factors regulating far-red light absorption of the red Chl *a*309/*a*303 pair, counterpart of the red Chl *a*305/*a*306 pair in euglenophytes and the red Chl *a*609/*a*603

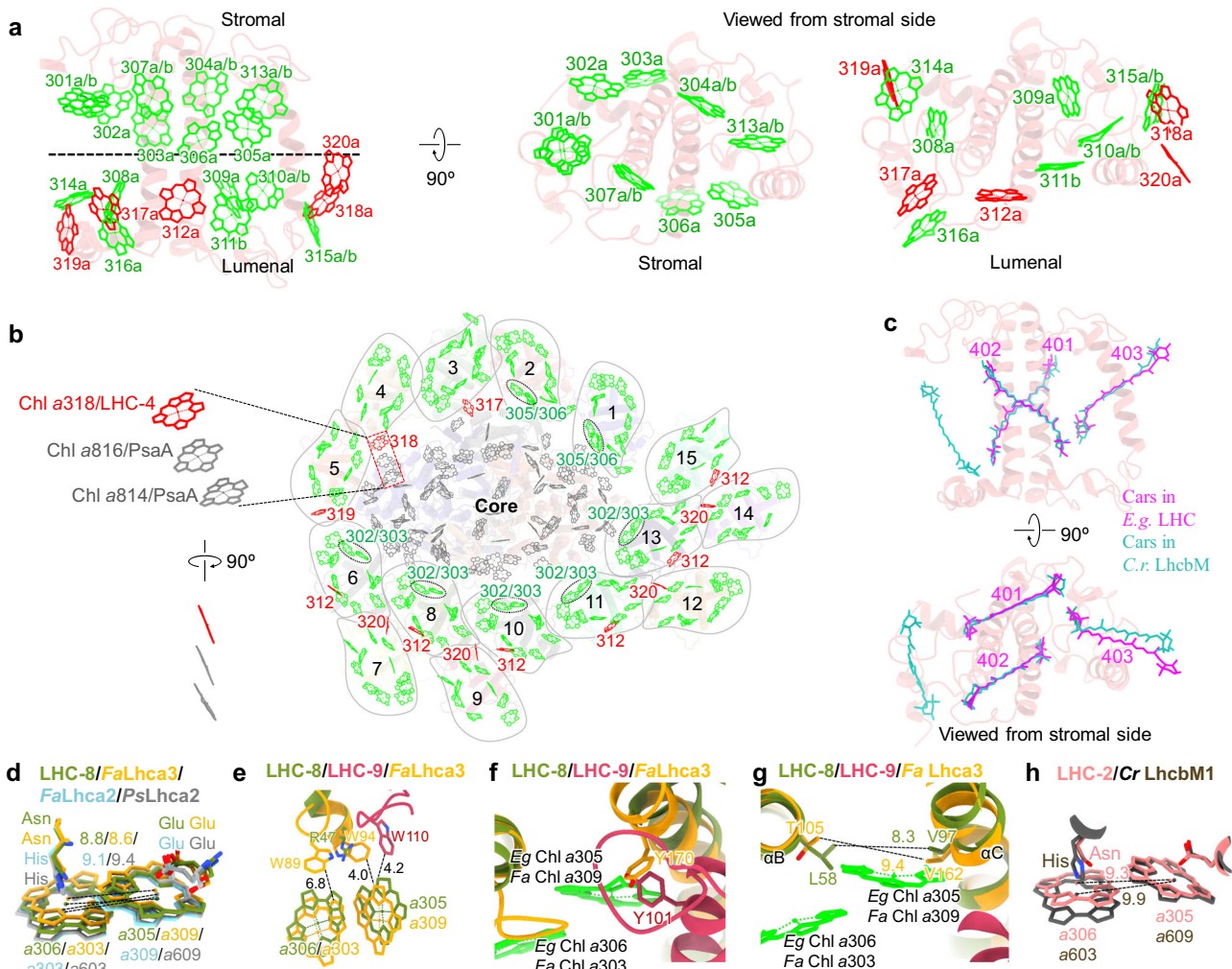

**Fig. 4 | The pigment binding sites in euglenophyte LHCs and Structural comparison of the local environment around Chl a305/a306 pairs in euglenophyte LHCs with their counterparts in *Fittonia albivenis* Lhca2/3, *Pisum sativum* Lhca2 and green algal LhcbM. a** Locations of the 20 Chl sites in euglenophyte LHCs. The stromal Chls and lumenal Chls are separated by dashed line (left panel, side view) and are viewed from the stromal side (right panel). Conserved Chl sites in green-lineage algae are colored green. Red Chls are specific to euglenophytes. Letters a and b represent Chl a and Chl b, respectively. **b** Arrangement of all Chls in euglenophyte LHCs. The core Chls are colored in gray. Specific Chls are colored in red. The parallel configuration of Chl 318/LHC-4, Chl 814/PsaA and Chl 816/PsaA are highlighted. The black ovals indicate the Chl a305/a306 pairs and Chl a302/a303 pairs. **c** Locations of 3 Car sites in euglenophyte *Euglena gracilis* (*Eg*) LHCs (megenta) and comparison with Car sites in *Chlamydomonas reinhardtii* (*Cr*) LhcbM (turquoise) viewed from the side (top) and stromal surface (bottom). **d** Comparison the red Chl pairs and their axial ligands in *E. gracilis* LHC-8, *Fittonia albivenis* Lhca2/Lhca3 (8WGH), and *Pisum sativum* Lhca2 (4XK8). The number labeled nearby the black dashed lines indicate the Mg-Mg distances in Å between Chls in Chl pairs. **e–g** Structural comparison of the local environment around Chl a305/a306 pair in LhcE dimers of euglenophyte *Euglena gracilis* (*Eg*) with their counterpart in *Fittonia albivenis* (*Fa*) Lhca3 (8WGH). The minimum distances between Trps and the Chl tetrapyrrole macrocycles in **e** and the nearest distances between αB and αC in **g** are indicated by black dashed lines with distances labeled in Å. **h** Comparison of axial ligand of Chl a306 in euglenophyte *Euglena gracilis* (*Eg*) LhcbM with that of its counterpart in *Chlamydomonas reinhardtii* (*Cr*) LhcbM (6KAD). The Mg-Mg distances between Chls in Chl pairs are indicated by black dashed lines with distances labeled in Å.

pairs in green algae and land plants, within the *Fittonia albivenis* LHCI (Lhca3)[50]. While Asn acts as the axial ligand of Chl a603, it induces stronger red-shifted absorption compared to His serving as the axial ligand[52,53]. As Asn has a shorter side chain than the His residue, the distance between Chl a603 and Chl a609 becomes smaller and their excitonic coupling strength might be increased as a result. In LHC-8/13, the axial ligand of Chl a306 is Asn, the same as that of its counterpart in *Fa* Lhca3 (Chl a303), whereas the axial ligands for its counterparts in *Fa* Lhca2 (Chl a303) and in *Pisum sativum* Lhca2 (Chl a603) are His residues (Fig. 4d). The distance between Chl a306 and Chl a305 of LHC-8/13 becomes smaller, in line with that of *Fa* Lhca3. Thus, their excitonic coupling strength might be increased, leading to enhanced far-red light absorption. Comparison with *Fa* Lhca3 has identified specific structural features potentially involved in regulating far-red light

absorption in euglenophyte LhcEs (Fig. 4e–g). Trp89 has been suggested to enhance far-red light absorption of red Chl pair (a303 and a309) in *Fa* Lhca3 due to its large side chain. Previous studies indicated that the Trp residues near Chls can induce distortion of the tetrapyrrole macrocycle, causing a red shift in Qy absorption band of Chls[51,54]. Except for Trp89, we found Trp94 is also located near the Chl a309 in *Fa* Lhca3, which may enhance far-red light absorption. In euglenophyte LhcE, the loop containing Trp89 is absent and Trp94 is replaced by Arg47 (Region 1) (Fig. 4e and Supplementary Fig. 7). However, the absence of Trp89 and Trp94 is compensated by a neighboring Trp110 (Region 1) in the opposing LhcE within the euglenophyte LhcE dimer (Fig. 4e and Supplementary Fig. 7). The distance from Trp110 to the tetrapyrrole macrocycle of red Chl a is comparable to those from *Fa* Trp89 and Trp94, suggesting that Trp110

may facilitate far-red light absorption of the red Chl pair in eugleno-phyte LhcE dimer. The polar Tyr170 surrounding Chl *a*309 in *Fa* Lhca3, which may favor its far-red light absorption, is replaced by Gly (Region 2) in euglenophyte LhcEs. Conversely, a Tyr (Region 2) in the opposing LhcE of the dimer, located at a similar position but with a different orientation, may serve a similar function (Fig. 4f). These results suggest that the Chl *a*305/*a*306 pairs in euglenophyte LhcEs have evolved a specific, dimer-dependent mechanism for regulating far-red light absorption.

Furthermore, the large distance (-10 Å) between Thr105 of αB and Val162 of αC in the neighborhood of Chl *a*309 in *Fa* Lhca3 has been suggested to contribute to the far-red light absorption[50]. The distance is much larger than the corresponding gap between Phe$_{αB}$ and Met$_{αC}$ of pea Lhca3 (-4 Å) and between Tyr$_{αB}$ and MetT$_{αC}$ of maize Lhca3 (-3 Å)[50]. The Thr-105$_{αB}$ and Val-162$_{αC}$ are conserved in Acanthaceae *F. albivenis* Lhca3 and two available Lhca3 sequences from Acanthaceae plants, *Strobilanthes cusia* and *Andrographis paniculata*, which exhib-ited a comparable red shift in fluorescence emission to *F. albivenis*. In addition, two non-Acanthaceae plants *Ananas comosus* and *Acer negundo* have the conserved residues as the three Acanthaceae plants. The fluorescence emission spectra of *A. comosus* and *A. negundo* have λ$_{max}$ values of 747.2 nm and 744.4 nm, respectively, indicating a red shift relative to other non-Acanthaceae plants[50]. The shortest distance between the corresponding residues in αB (Leu/Met) and αC (Ser/Ala/Ile/Val) (Supplementary Fig. 7, Region 3) of euglenophyte LhcEs is approximately 8 Å (Fig. 4g), slightly smaller than that of *F. albivenis* Lhca3 and notably larger than those of pea and maize. Therefore, the large gap between αB and αC may enhance far-red light absorption of red Chl *a*305/*a*306 pair in euglenophyte LhcEs. The large gap between αB and αC results from residues with small side chains, which provide a smaller interaction surface with the red Chl compared to residues with larger side chains. This difference may affect the conformation of Chl and thereby alter its light-harvesting properties. The Chl *a*302/*a*303/*a*304 cluster is equivalent to the Chl *a*610/*a*611/*a*612 clusters found in land plants and green algae, which are low-energy red-shifted cluster in LHCs and represent the lowest-energy site in LHCII[55,56]. Due to the specific orientation of euglenophyte LHC-6/8/10/11/13, the low-energy red Chl *a*302/*a*303/*a*304 cluster, situate in the interfacial region of LHCI and the PSI core (Fig. 4b). Similarly, the Chl *a*305/*a*306 pairs of LhcbM type LHC-2 is located at the interfacial region of LHC and the PSI core, distinct with the corresponding Chl *a*609/*a*603 pairs in LhcbM trimers of green algae and land plants (Fig. 4b). The interac-tions between LHC and the PSI core may affect the conformation of the red Chl pairs positioned at the interface and facilitate far-red light absorption. This is supported by the enhancement of the red shift of LHC when it binds to the PSI core[50,57]. Moreover, the axial ligands of Chl *a*306 in LhcbMs (LHC-2/3) are Asn instead of His in green algal LhcbMs, which may enhance the far-red light absorption (Fig. 4h).

In summary, based on the distinct architectures and pigment arrangements of their light-harvesting complexes, euglenophyte LHCs may adopt alternative structural features to enhance far-red light absorption compared to LHCs of green algae and land plants, high-lighting a distinct evolutionary strategy of euglenophytes for light adaptation during secondary endosymbiosis.

### Energy transfer within euglenophyte PSI−LhcE−LhcbM

The simplified PSI core and the specific arrangement of LHCs construct a specific pigment network. We performed molecular mechanics force field optimization on the entire system using Amber22[58]. Using the optimized structural data, we performed computational simulations to estimate excitation energy transfer (EET) rates (time constants) between all Chl pairs within euglenophyte PSI−LhcE−LhcbM, accord-ing to the Förster theory[59] (Fig. 5a). Due to the limited map resolution, the assignment of Chl types at certain low-resolution sites is not unambiguously. Thus, the simulations based on the current structure

may contain uncertainties. Quantum simulations reveal that EET between Chls occurred on a picosecond or subpicosecond time scale, indicating efficient EET within the euglenophyte PSI−LhcE−LhcbM.

To further probe the system, we employed the generalized För-ster (GF) theory, an extension of the classical Förster theory, to eval-uate the EET rates between LHCs and between LHCs and the PSI core (Fig. 5b)[60]. Our analysis reveals that the EET pathways from LHCs to the PSI core can be divided into three groups: LHC-1 (Group I), LHC-2/3/4/5 (Group II), and LHC-(6-15) (Group III), with relatively low EET rates between the groups due to the large gap between the Chls at their interface (Fig. 5b). In group I, LHC-1 efficiently transfer energy to the PSI core through the stromal Chl 305/306 pair and lumenal Chl 311 (Fig. 5c, d). In group II, direct energy transfer from LHC-2 to the PSI core is less efficient because of the large distance between Chls at the interface. However, LHC-2 can efficiently transfer energy to LHC-3 through stromal Chl 301/307, with subsequent transfer to the PSI core via stromal Chl 301 and lumenal Chl 317 of LHC-3. The specific Chl 317 forms multiple EET pathways with adjacent Chls, indicating its critical role in mediating energy transfer (Fig. 5d). The specific Chl 318 is also crucial for the EET from the antenna to the PSI core. The parallelly arranged Chl 318$_{LHC-4}$, Chl 816 $_{PsaA}$ and Chl 814$_{PsaA}$ mediate energy transfer from LHC-4 to the PSI core on a subpicosecond time scale (-0.1 ps). The EET rate between LHC-4 and LHC-5 of LHC dimer is fast within two picoseconds, mediated by the Chl 305/306 pairs (Fig. 5c). Consequently, LHC-5 transfers energy to the PSI core mainly through LHC-4 (Fig. 5b). In addition, LHC-5 can also transfer energy directly to the PSI core through Chl 301 and Chl 305/306 pair. Efficient EET between the LHC-2/3 dimer and the LHC-4/5 dimer is mediated by stomal Chl 302/303 pairs and lumenal Chl 309/314.

In group III, EET from the inner LHC-6/8/10/13 to the PSI core is facilitated by stromal Chl 302/303 pairs and lumenal Chl 314. This mechanism differs from those in green algal PSI−LHCIs mediated by Chl 305/306 pairs and lumenal Chl 311, resulting from their opposite orientation relative to green algal LHCIs[4–12]. Highly efficient EET within the LHC-6/7 dimer, LHC-8/9 dimer, LHC-11/12 dimer, and LHC-13/14 dimer is mediated by the Chl 305/306 pairs, similar to the LHC-4/5 dimer (Fig. 5c). Additionally, the specific Chl 320 facilitate the EET within LHC dimers in group III. Thus, EET from the outer LHC-7/9/12/14 is funneled to the PSI core by LHC-6/8/11/13, respectively. EET from LHC-8 to the PSI core is relatively slow, whereas efficient EET occurs between LHC-8 and LHC-6 via Chl 301/308/312/314$_{LHC-8}$ and Chl 304/309/313/320$_{LHC-6}$ and between LHC-8 and LHC-10 via Chl 309/313/320$_{LHC-8}$ and Chl 301/308/312/314$_{LHC-10}$, resulting in two additional EET pathways from LHC-9 to the PSI core: 9-8-6-core (22.6 ps) and 9-8-10-core (14.1 ps). In contrast, EET from LHC-11/15 to the PSI core appeared less efficient due to the greater distances between the Chls of LHC-11/15 and the PSI core (Fig. 5b). On the other hand, LHC-11 can efficiently transfer energy to Chl 304/309/313 of adjacent LHC-10 via Chl 301/314 and to Chl 301/314 of adjacent LHC-13 via Chl 304/309/313. Therefore, EET from the outer LHC-12 to the PSI core are mainly through two routes: 12-11-10-core (9.9 ps) and 12-11-13-core (8.7 ps). Similarly, LHC-15 transfers energy to Chl 304/309/313/320 of adjacent LHC-13 via Chl 301/308/312/314 before reaching the PSI core. Notably, the specific Chl 320 establishes abundant EET pathways with Chls of adjacent LHCs, including Chl 305/306 pairs, Chl 307, Chl 308, Chl 309, Chl 312, and Chl 313, indicating its critical role in mediating energy transfer.

In summary, our analysis suggests that the euglenophyte PSI−LhcE−LhcbM supercomplex is optimized for efficient energy transfer from its peripheral antennae to the PSI core. This efficiency is underpinned by the specific EET pathways shaped by the simplified PSI core and the specific arrangement of LHCs. The Chl 305/306 pairs, Chl 302/303 pairs, Chl 301/304/308/309/311/313/314, and specific Chl 312/317/318/320 play critical roles in mediating the EET from LHCs to the PSI core. The presence of these specific Chls aligns with the distinct organizational pattern of LHCs in euglenophyte PSI−LhcE−LhcbM,

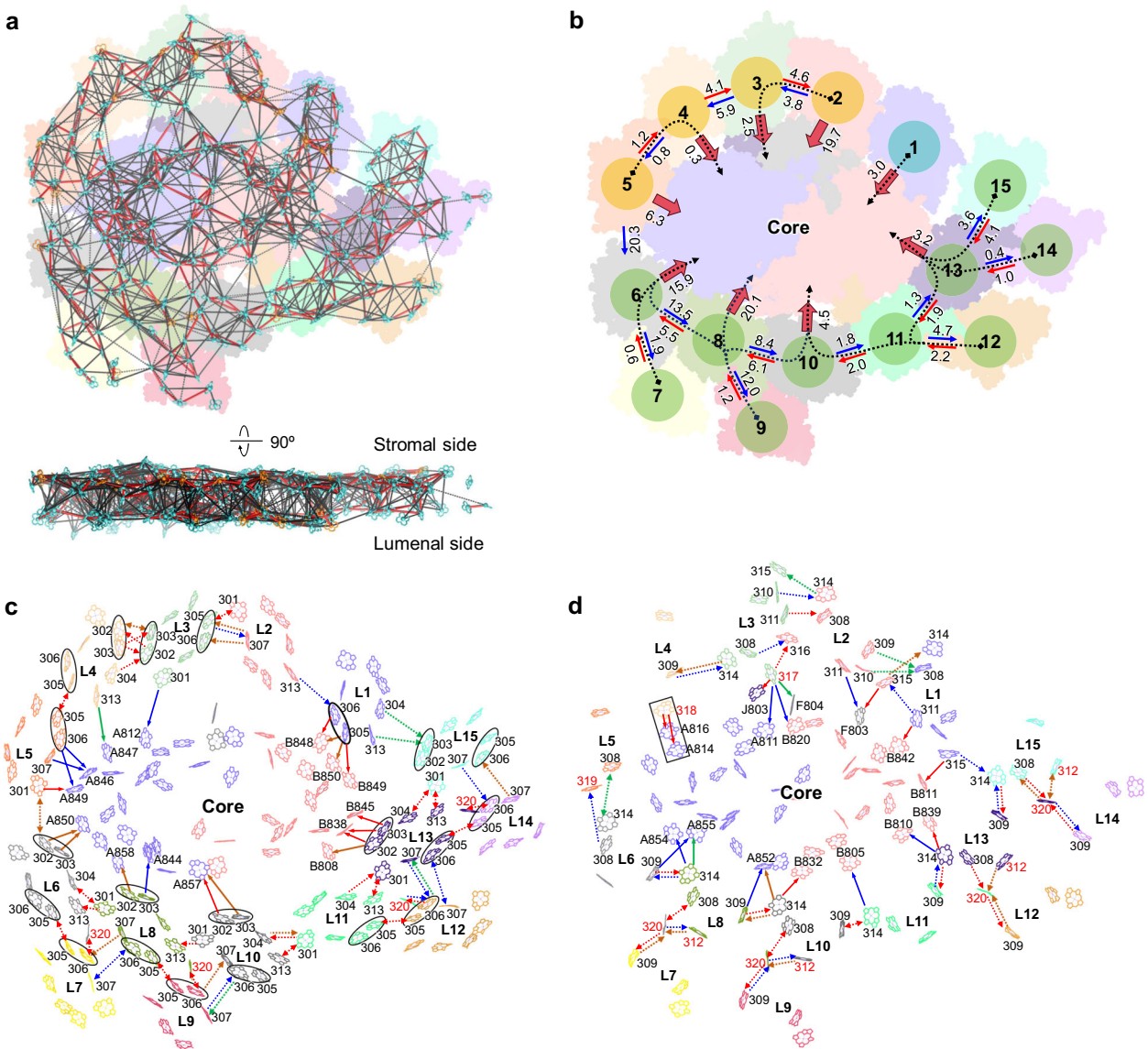

**Fig. 5 | Excitation energy transfer pathways in euglenophyte PSI–LhcE–LhcbM.**
**a** Stomal view (top) and side view (bottom) of the maps of interpigment EET rates based on the Förster theory. The lines indicate rates faster than 1 ps (red), in the 1-10 ps range (black solid line) and in the 10-20 ps range (black dashed line). Inter-pigment ratERTes slower than 20 ps are omitted. **b** EET rates labeled in ps between LHCs (blue and red arrows) and from LHCs to the PSI core (thick red arrows) based on the generalized Förster theory. Rate longer than 25 ps are omitted. The relatively efficient EET pathways from LHCs to PSI core are indicated by dashed arrows. The LHCs are divided into three group with relatively low EET rates between them

(group I: turquoise, group II: orange, and group III: green). EET between the Chls at the stromal layer (**c**) and lumenal layer (**d**) viewed from the stromal side. EET pathways from LHCs to PSI core (solid arrows), between LHCs (dashed arrows). Red arrows: faster than 2 ps; brown arrows: between 2 ps and 5 ps; blue arrows: between 5 ps and 10 ps, green arrows: between 10 and 20 ps. The black ovals indicate the Chl 305/306 pairs and Chl 302/303 pairs. The black square indicates the parallelly arranged Chl 318$_{LHCl-4}$, Chl 816 $_{PsaA}$ and Chl 814$_{PsaA}$. The specific Chl-binding sites are colored red. L1-L15 indicate 15 LHCs around the PSI core.

enabling efficient and robust energy transfer within the giant pigment network of the photosynthetic supercomplex.

## Insights into the evolution of green-lineage PSI–LHCs
Viridiplantae originated from cyanobacteria through primary endosymbiosis, subsequently diversifying into two major lineages: the Chlorophyta (green algae) and Streptophyta (Charophyta and land plants)[61]. Both green algal and Streptophyta PSI–LHCIs share a nearly identical PSI core and Lhca antenna "belt", supporting their derivation from a common ancestral PSI–LHCI. The ancestral PSI was derived from the trimeric PSI of cyanobacteria and was transitioned to a monomeric form owing to the conformational change of the

C-terminus of PsaL[4,62]. The integration of membrane-embedded LHCIs with PSI eventually led to the construction of PSI–LHCI supercomplexes (Fig. 6).

Green algae include prasinophytes and core chlorophytes. Prasinophytes are near the basal position in the evolution of green algae and are paraphyletic group, including Mamiellophyceae, Pyramimonadophyceae, Nephroselmidophyceae, etc[63]. Core chlorophytes, which have evolved from a branch of prasinophytes, form a monophyletic group with greater morphological and ecological diversity. The structural similarities observed in the PSI core and antenna organization among these groups underscore their close evolutionary relationship (Fig. 6). Notably, the presence of

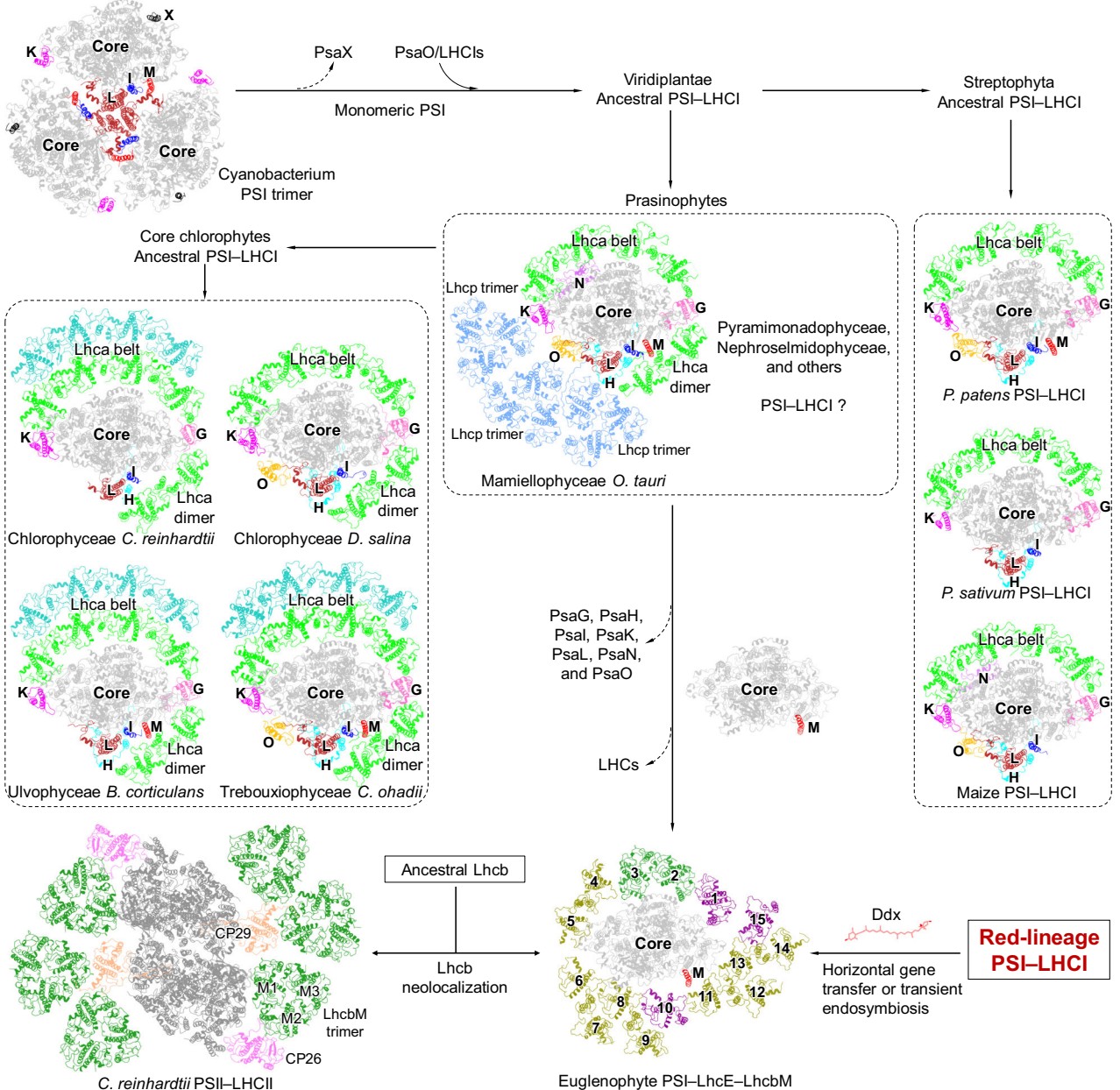

**Fig. 6 | Possible evolutionary development of green-lineage PSI–LHC super-complexes.** PDB codes of PSI structures: trimeric PSI of cyanobacterium *Syne-chococcus elongatus* (1JBO); PSI–LHC of a red alga (*Cyanidium caldarium*, 8WEY); PSI–LHCI–Lhcp of green alga *Ostreococcus tauri* (7YCA); PSI–LHCI of green alga *Chlamydomonas reinhardtii* (6IJO), *Dunaliella Salina* (6SL5), *Bryopsis corticulans* (6IGZ), and *Chlorella ohadii* (7A4P); PSI–LHCI of land plants *Physcomitrella patens* (7KSQ), *Pisum sativum* (4XK8), and Maize (5ZJI); PSII–LHCII of *C. reinhardtii* (6KAD). Letters G, H, I, K, L, N, O, and X indicate PsaG, PsaH, PsaI, PsaK, PsaL, PsaN, PsaO, and PsaX, respectively.

prasinophyte-specific Lhcp trimers in the *O. tauri* PSI–LHCI–Lhcp supercomplex may reflect ancestral features of green algal PSI–LHCI. Subsequently, prasinophytes and core chlorophytes were acquired by non-photosynthetic eukaryotes via secondary endosymbiosis, leading to the emergence of Euglenophyta and Chlorarachniophyta, respectively[64]. In particular, Euglenophyta originated from an evolutionary event when a green alga closely related to the extant *Pyramimonas parkeae* was engulfed and enslaved by a phagotrophic euglenid[31].

Our structural analysis reveals significant variations between euglenophyte PSI–LHC and green algal PSI–LHCs, in terms of the PSI core composition, LHC composition and arrangement, and pigment association. During endosymbiosis, the green algal PSI core underwent substantial simplification, losing several subunits, including PsaG,

PsaH, PsaI, PsaK, PsaL, PsaN, and PsaO, consequently forming the most simplified PSI core of euglenophyte known at the genetic level (Fig. 6). The highly conserved PsaD subunit undergone conformational changes, facilitating LHC binding. Structural and phylogenetic analyses reveal a close relationship between euglenophyte LHCs and green algal Lhcbs (Supplementary Fig. 8 and Supplementary Fig. 10), suggesting that euglenophyte LHCs may have evolved from green algal Lhcbs. Based on the structural and sequence similarities, the euglenophyte LhcbM dimer is speculated to have evolved from the green algal LhcbM trimer. Other euglenophyte LHCs may originate from green algal Lhcb through LHC "neolocalization", an evolutionary phenomenon defined as the structural relocalization and modifications of LHCs[20,65]. LhcE13 (LHC-1) possesses several structural features reminiscent of LhcbM, including the BE loop, CE loop, and the Neo binding

site (Supplementary Fig. 10a), although its N-terminal domain differs from those of LhcbMs and more closely resembles those of LhcEs (LHCI-4 to LHCI-15) (Fig. 2c). Phylogenetic analysis also shows that LhcE13 diverged earlier than other LhcEs (Supplementary Fig. 8)[37], suggesting that LhcE13 may represent an evolutionary intermediate between LhcbM and other LhcEs during LHC neolocalization.

Another notable distinction is the pigment composition. Euglenophyte LHCs predominantly contain Ddx, a carotenoid type absents from green algal LHCs, which instead are rich in lutein and violaxanthin, the pigments that are not identified in euglenophytes[4–7,41,66]. Ddx was identified exclusively in red-lineage LHCs[23–27], and the presence of red-lineage genes in the *E. gracilis* nuclear genome, acquired via horizontal gene transfer or transient endosymbiosis[32], suggests that the Ddx in euglenophyte LHCs is derived from red-lineage algae.

Overall, our findings demonstrate that euglenophyte PSI–LhcE–LhcbM has undergone a complex evolutionary pathway, resulting in a chimeric structure that incorporates elements from both green and red algal lineages.

## Methods

### Purification of PSI–LhcE–LhcbM from *Euglena gracilis*

*Euglena gracilis* (catalog number: GY-D32) was purchased from Shanghai Guangyu Biological Technology Co., Ltd and cultured in AF-6 medium with constant air-bubbling at 23 °C under continuous light at 40 μmol photon m$^{-2}$ s$^{-1}$. Cells were harvested by centrifugation (6000 × $g$, 8 min) and washed twice with buffer 1 (25 mM MES-NaOH, pH 6.5). Subsequently, cell disruption was performed in buffer 2 (25 mM MES-NaOH, pH 6.5, 1.0 M betaine, 10 mM MgCl$_2$) using glass beads. Unbroken cells and debris were removed by low-speed centrifugation (1000 × $g$, 2 min). The thylakoid membranes in supernatant were collected by centrifugation at 21,000 × $g$ for 20 min, and then washed with buffer 3 (25 mM MES-NaOH, pH 6.5, 1.0 M betaine, 1.0 mM EDTA). The thylakoid membranes were solubilized in buffer 4 (25 mM MES-NaOH, pH 6.5, 1.0 M betaine, 10 mM NaCl, 5.0 mM CaCl$_2$) using 2.5% (w/v) n-dodecyl-α-D-maltopyranoside (α-DDM from Anatrace, USA) at 0.4 mg mL$^{-1}$ Chl on ice for 15–20 min. After clarification by centrifugation (21,000 × $g$, 20 min), the supernatant was loaded onto a discontinuous sucrose density gradient (10–30% sucrose, 11 concentration gradients, with the interval between each gradient being 2%) containing 0.02% α-DDM. Centrifugation was performed at 230,500 × $g$ for 20 hours using an SW40Ti rotor (Beckman, USA). The PSI–LHC band was collected, concentrated to a Chl concentration of 1.5 mg mL$^{-1}$ using a 100 kDa molecular weight cutoff filter (Amicon Ultra; Millipore), and stored at 4 °C.

### Biochemical characterization of *E. gracilis* PSI–LhcE–LhcbM

Absorption spectra were recorded at room temperature using a Shimadzu UV–Vis 1900 spectrophotometer. The PSI–LhcE–LhcbM subunit composition was analyzed by SDS-PAGE using 8–16% gradient gels and identified by mass spectrometry analysis. The protein bands were cut out from the gel, reduced with dithiothreitol, alkylated with iodoacetamide, and digested with trypsin. Resulting peptides were analyzed by liquid chromatography-tandem mass spectrometry (LC-MS/MS) using an Easy-nLC 1000 system (Thermo Fisher) coupled to a Q Exactive mass spectrometer (Thermo Fisher). Peptides were separated using a phase trap column (nanoViper C18, 100 μm × 2 cm, Thermo Fisher Scientific) connected to a C18-reversed phase analytical column (75 μm × 10 cm, 3 μm resin, Thermo Fisher Scientific). The acquired spectra were searched against the selected database using MASCOT engine (version 2.4) with the Proteome Discovery searching algorithm (version 1.4) (Supplementary Data 1).

Pigments were extracted by pre-cooled 100% acetone overnight at 4 °C protected from light. Pigment composition was analyzed by high-performance liquid chromatography (Shimadzu, Japan) using a C18 reversed-phase column (Waters, Ireland) with a Shimadzu photodiode array detector following previously described procedures[25]. The elutes were detected by a Shimadzu photodiode array detector at 445 nm with a wavelength detection range of 300–800 nm. Five pigments were identified based on the characteristic absorption peaks of their absorption spectra and elution profiles, in line with previous reports[67,68].

### Sequence analysis of *E. gracilis* PSI–LhcE–LhcbM

Transcriptome sequencing of *E. gracilis* was performed by Huada using high-throughput Illumina technology. Total RNA was extracted from *E. gracilis* cells, and a cDNA library was constructed using the NEBNext Ultra RNA Library Prep Kit for Illumina (NEB, MA, USA). First-strand cDNA synthesis utilized random hexamer primers, followed by second-strand synthesis with RNase H and DNA polymerase I. The double-stranded cDNA underwent end repair, A-tailing, and adapter ligation. cDNA fragments of approximately 150 bp in length were selected using the AMPure XP system (Beckman Coulter, Beverly, USA). To obtain the final cDNA library, cDNA was first treated with USER Enzyme (NEB) and then amplified using PCR. High-throughput sequencing was performed using paired-end reads on a Huada DNBSEQ platform. De novo transcriptome assembly was performed using Trinity with the paired-end reads (left.fq and right.fq).

Sequences encoding the *E. gracilis* PSI core subunits and LHCs were identified by querying this transcriptome assembly using homologous sequences. Sequence alignments were generated using CLC Sequence Viewer 8.0 and visualized with ESPript 3.0. For phylogenetic analysis, homologous sequences were aligned using MUSCLE (default parameters). Phylogenetic trees were constructed in MEGA X using the Neighbor-Joining method. Evolutionary distances were calculated using the Poisson correction method (units: amino acid substitutions per site), incorporating pairwise deletion to handle ambiguous positions. Bootstrap support values, based on 1000 replicates, are indicated at branch nodes[69]. The LHC sequences of *Chlamydomonas reinhardtii*[70] and *Arabidopsis thaliana*[71] for phylogenetic analysis were retrieved from the Phytozome database (https://phytozome.jgi.doe.gov/)[72].

### Cryo-EM data collection and processing

*E. gracilis* PSI–LhcE–LhcbM sample (2.5 μL, 2.0 mg mL$^{-1}$ Chl) was applied onto the glow-discharged grids (Quantifoil Au R2/1, 200 mesh) for blotting using a Vitrobot Mark IV (Thermo Fisher Scientific) with 100% humidity and 4 °C. Data collection was performed on a 300 kV Titan Krios G3i microscope (Thermo Fisher Scientific) equipped with a K3 BioQuantum direct electron detector (Gatan Inc.) and a GIF Quantum energy filter (Gatan). Images were acquired in super-resolution mode at a nominal magnification of ×81,000, yielding a calibrated pixel size of 0.53 Å. A total of 5007 movie stacks were recorded with a defocus range of −1.2 to −2.2 μm and a total accumulated dose of 50 e⁻ Å$^{-2}$, using a 20 eV energy slit.

Image processing was mainly performed by cryoSPARC 3.3.1[73]. Movie stacks were patch-motion corrected and dose-weighted, followed by binning by a factor of 2. Contrast transfer function (CTF) parameters were estimated for each micrograph using Patch CTF. After particle-extraction and two rounds of reference-free 2D classifications, 131,470 particles were selected with obvious incorrectly selected particle positions excluded from the particle set for Ab-initial reconstruction and 3D classification. A final set with 78,124 particles were subjected to homogeneous refinement. To improve the resolution of the density map, after 3D non-uniform refinement and sharpening, global (per-group) CTF refinement, local (per-particle) CTF refinement and particle subtraction followed by local refinement with soft mask for the regions of the peripheral LHCs were performed using cryoSPARC. The final overall resolution of the refined map, determined using the gold-standard Fourier Shell Correlation (FSC) criterion at 0.143, was 2.72 Å. The local resolution in the peripheral LHCIs region reached 3.29 Å.

## Model building and refinement

To build the model of *E. gracilis* PSI–LhcE–LhcbM supercomplex, the PSI core of green algal PSI–LHCI (PDB ID: 6IJO) was fitted into a 2.72 Å overall cryo-EM map using UCSF Chimera[5,74]. The amino acid residues of each chain were subsequently corrected by referring to the sequence of the counterpart in *E. gracilis* from transcriptome sequencing using COOT[75,76]. For the LHCs, LhcbM of *Chlamydomonas reinhardtii* PSI–LHCII (PDB ID: 6KAD) was fitted to the map. The potential sequences of LHCs were searched from the transcriptome by blast using sequence of LhcbM. The sequence of each LHC was assigned based on the best match between the amino acid sequence and the cryo-EM map. The assigned sequence of each LHC was listed in Supplementary Table 5 with the corresponding accession number to each sequence in the National Center for Biotechnology Information. LHC proteins of LhcE5/6/7/8/10/11/13 families and LhcbM2/M8 families were identified, in line with the mass spectrometry analysis in recent study[37]. Due to the resolution limitation of the density map of LHC-12/13/14/15 in unit 11-15, their sequence could not be assigned unambiguously. The unit 11-15 share similar structural features with unit 6-10, so we tentatively used the sequences and structures of LHC-7/8/9/10 to build the model of LHC-12/13/14/15 respectively. For the assignment of pigments, most of the Chl *b* molecules in the structure were tentatively assigned when the density for the C7-formyl group is present. The assignment of Chl *b*311 in LHC-1/2/3 was further supported by hydrogen bonding of the C7-formyl group with charged residues or polar residues; the assignment of Chl *b*313/*b*315 in LHC-1/2, Chl *b*301 in LHC-2, and Chl *b*313 in LHC-3 were further supported by sequence conservation of the binding site relative to the binding sequences of their counterparts in green algal LhcbM. The density map in these regions lacks sufficient resolution for the assignment of Chl *b*301 in LHC-4/7/12/15. Nevertheless, because the binding sequence of Chl *b*301 in LHC-4/7/12/15 is conserved compared to that of Chl *b*301 in LHC-5/6/10, Chl *b*301 was tentatively assigned in LHC-4/7/12/15. Moreover, the assignment of Chls *b* in LHC-1/2/3 was also referred to their counterparts in green algal LhcbM (Supplementary Table 6). Ddx molecules were discriminated based on the density of the Car head group. Neoxanthin was identified based on its hook-shaped density. *E. gracilis* contained 5′-hydroxyphylloquinone as the dominant naphthoquinone of PSI. Phylloquinone is present in minimal quantities in *E. gracilis* PSI, and it was estimated that only one 5′-hydroxyphylloquinone is bound to the *E. gracilis* PSI[77,78]. Due to the limited map resolution and the high structure similarity between 5′-hydroxyphylloquinone and phylloquinone, these two naphthoquinones could not be distinguished explicitly based on the density map. Therefore, the naphthoquinones in the current structure were tentatively assigned as phylloquinone, as the map density corresponding to the hydroxyl group at the 5′ position of 5′-hydroxyphylloquinone is absent. The Grade Web Server was used to generate geometrical restraints of pigments. All residues and cofactors were manually adjusted using the COOT software. The constructed model was refined using Phenix real-space refinement, incorporating geometry and secondary structure restraints[79,80]. The final atomic model was obtained through iterative cycles of manual correction and Phenix refinement. The geometries of the structural models were evaluated using Phenix (Supplementary Table 1).

## Simulation of the excitation energy transfer in *E. gracilis* PSI–LhcE–LhcbM

Molecular mechanics (MM) force field optimization was conducted using Amber22[58]. An initial structural relaxation involving 50,000 steps of energy minimization was performed on the entire system, during which positional restraints on the protein were gradually released. The minimization process employed an alternating scheme of the steepest descent and conjugate gradient methods, with

transitions every 1000 steps. The FF14SB force field[81] was applied to the protein, whereas parameters for Chls were adopted from prior parameterizations of bacteriochlorophyll *a*[82].

The first excited-state energies of individual pigments were computed independently using time-dependent density functional theory (TDDFT)[83] as implemented in Gaussian16 (Gaussian, Inc. Wallingford, CT, USA) at the CAM-B3LYP/6-31 G* level of theory[84]. Subsequently, pairwise excitation energy transfer time constants between pigments were estimated using the transition charge from the electrostatic potential (TrEsp) method[85] via the Multiwfn program[86], with customized Python scripts (https://doi.org/10.5281/zenodo.10791187). These calculations were based on Förster resonance energy transfer (FRET) theory[59]. Furthermore, cluster-level excitation energy transfer time constants were computed using a generalized Förster framework[60], leveraging the same computational tools and scripting infrastructure. The overall computational protocol closely followed that of our previous studies[23,27].

## Reporting summary

Further information on research design is available in the Nature Portfolio Reporting Summary linked to this article.

## Data availability

The cryo-EM map and atomic coordinates generated in this study have been deposited in the Protein Data Bank and Electron Microscopy Data Bank under the accession numbers 9VJS and EMD-65121, respectively. The RNA-seq data generated in this study have been deposited in the NCBI Sequence Read Archive (SRA) database under accession code PRJNA1420364. Source data are provided with this paper.

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

## Acknowledgements

We thank Qiu-Yao Jiang (Cryo-Electron Microscopy Platform of Medical Science and Technology Innovation Center of Shandong First Medical University), Dian-Li Zhao, Xiao-Wei Li, and Ceng Gao (Laoshan Laboratory, Qingdao, Shandong, China) and Lei Qi (Biomedical Research Center for Structural Analysis, Shandong University) for their contributions in cryo-EM data collection. We thank Xiao-Ju Li and Hai-Xia Zhu (State Key Laboratory of Microbial Technology, Shandong University, Qingdao, China) for their help with TEM. Numerical computations were performed on Hefei advanced computing center. This work was supported by National Key R&D Program of China (2023YFA0914600 to L.S.Z. and L.N.L.), Shandong Province Science Fund for Excellent Young Scholars (ZR2024YQ027 to L.S.Z.), National Natural Science Foundation of China (32570122 to L.S.Z., 32330001 to Y.Z.Z., W2441012 to Y.Z.Z., 32070109 to L.N.L., 21873034 to J.G.), National Key R&D Program of China (2021YFA0909600 to L.N.L.), the Taishan Scholars Program of Shandong Province, China (tsqn202306092 to Y.Z.Z.), Natural Science Foundation of Shandong Province (ZR2024QC042 to K.L.), Marine S&T Fund of Shandong Province for Qingdao Marine Science and Technology Center (2022QNLM030004-3 to Y.Z.Z.), the SKLMT Frontiers and Challenges Project (SKLMTFCP-2023-06 to Y.Z.Z.), Qilu Youth Scholar Startup Funding of SDU (to L.S.Z.), the Royal Society (URF\R\180030 to LNL), Biotechnology and Biological Sciences Research Council (BBSRC) (BB/Y01135X/1, BB/V009729/1, and BB/W001012/1 to L.N.L.), Fundamental Research Funds for the Central Universities (2662024XXPY003).

## Author contributions

Y.-Z.Z., L.-S.Z., and L.-N.L. conceived the project; B.-Y.Q., L.-S.Z., K.L., and H.-J.W. performed sample preparation and measurements. K.L. collected and processed the cryo-EM data. L.-S.Z. and B.-Y.Q. built the structural model and refined the structure. Q.W. and J.G. performed computational simulations for excitation energy transfer. L.-S.Z., K.L., B.-Y.Q., X.-X.Q., F. Z., and X.-L.C. analyzed the data. L.-S.Z., B.-Y.Q., L.-N.L, Y.-Z.Z., and X.-L.C. wrote the manuscript, with contributions from all other authors.

## Competing interests

The authors declare no competing interests.
