## [Transparent Peer Review file · Nature Communications]

Unique structure and energy transfer of a far-red-absorbing euglenophyte PSI–LhcE–LhcbM supercomplex

Corresponding Author: Professor Long-sheng Zhao

Version 0:

Reviewer comments:

Reviewer #1

(Remarks to the Author)

the authors did a good job answering my comments and i support the publication of the ms in its present form.

Reviewer #2

(Remarks to the Author)

The authors have adequately addressed my comments, and the manuscript has been greatly improved. I have no further questions and recommend publication of this manuscript in Nature Communications.

Reviewer #3

(Remarks to the Author)

The authors, in this revised version, present evidence that the two polypeptides they designate as LhcbM2 and LhcbM8 share a common origin with green algal LhcbM proteins and have diversified independently from the other LhcbMs that function as major LHCII. In the recently published structural analysis of the *Euglena* PSI–LHCI supercomplex by another group (Kato et al., 2025, *Sci. Adv.*), this issue was not explored in depth, and all associated LHCs were simply designated as LHCI-1 through LHCI-13. In contrast, the authors of the present manuscript appear to place greater emphasis on the association of LhcbM2/LhcbM8. As long as they refrain from using terminology such as “PSI–LHCI–LHCII supercomplex,” as in the previous version, I do not intend to raise further objections on this point. After all, *Euglena* is a secondary endosymbiont of the green lineage; the incorporated LhcbM-type antenna proteins were likely used initially only in the PSII light-harvesting system but may have been gradually co-opted into the PSI antenna system over the time. I would, however, like to point out one issue: the evidence supporting the claim that the sequences of LhcbM2/LhcbM8 are closely related to the green algal LhcbM group is extremely weak. According to Fig. S8, the support value for the branching node separating LhcbM2/8 from *Euglena* LHCI is only 59%. Moreover, the paired protein is not LhcbM5 but the monomeric Lhcb5 (CP26). This can only be regarded as very weak support. There also appears to be an argument (Fig. 2D) implying trimeric organization of LhcbM2 based on the designation “LhcbM,” but such reasoning does not make sense.

Reviewer #4

(Remarks to the Author)

This manuscript reports the determination of the structure of a PSI–LHCI–LHCII supercomplex of *Euglena gracilis* at 2.72 Å resolution using cryo-electron microscopy and presents the calculation of excitation energy transfer pathways within the supercomplex. The data presented in the present study are sound and the descriptions in the manuscript are clear. The supercomplex contained a “minimal” PSI core consisting of eight subunits and 15 LHC complexes. The minimal PSI core and the nomenclature of 15 LHCs (LHCBMs and LHCEs) were reported by Miranda-Astudillo et al. (*Experimental Biology*, 2025), which the authors cite as ref. 37. Recently, the structure of the PSI-LHC supercomplex of *Euglena gracilis* has been determined at a resolution of 2.82 Å by Kato et al (2025), as cited as ref 38 in this manuscript. This supercomplex contains

the minimal PSI core and 13 LHC2. The structure is very similar but lacks two LHCS (LHC14/15). The structure determined in the present study is basically similar to that reported in ref 38. Thus, the most important finding on the structure of the supercomplex in the present manuscript is that two additional LHCS (LHCE5/8) were determined in the supercomplex structure. The significance of the present study is somewhat limited.

Specific points

L121 minimal PSI configuration: the authors use “minimal” and “simplified” to describe the PSI core consisting of eight subunits. It is better to use only one of them to prevent confusion for readers.

L149 “LhcbMX1-5” should be corrected as LhcbMX1-3 and LhcbMX5

L239 “The PsaJ and PsaF subunits, which occupy the position of LhcbM observed in the green algal LHCII trimer” is correct?

Responses to Reviewers' Comments

Reviewer 1:

The authors did a good job answering my comments and i support the publication of the ms in its present form.

Reply: We sincerely appreciate the reviewer for endorsing publication.

Reviewer 2:

The authors have adequately addressed my comments, and the manuscript has been greatly improved. I have no further questions and recommend publication of this manuscript in Nature Communications.

Reply: We sincerely appreciate the reviewer for endorsing publication.

Reviewer 3:

The authors, in this revised version, present evidence that the two polypeptides they designate as LhcbM2 and LhcbM8 share a common origin with green algal LhcbM proteins and have diversified independently from the other LhcbMs that function as major LHCI. In the recently published structural analysis of the Euglena PSI–LHCI supercomplex by another group (Kato et al., 2025, Sci. Adv.), this issue was not explored in depth, and all associated LHCs were simply designated as LHCI-1 through LHCI-13. In contrast, the authors of the present manuscript appear to place greater emphasis on the association of LhcbM2/LhcbM8. As long as they refrain from using terminology such as “PSI–LHCI–LHCII supercomplex,” as in the previous version, I do not intend to raise further objections on this point. After all, Euglena is a secondary endosymbiont of the green lineage; the incorporated LhcbM-type antenna proteins were likely used initially only in the PSII light-harvesting system but may have been gradually co-opted into the PSI antenna system over the time.

I would, however, like to point out one issue: the evidence supporting the claim that the sequences of LhcbM2/LhcbM8 are closely related to the green algal LhcbM group is extremely weak. According to Fig. S8, the support value for the branching node separating LhcbM2/8 from Euglena LHCI is only 59%. Moreover, the paired protein is not LhcbM5 but the monomeric Lhcb5 (CP26). This can only be regarded as very weak support. There also appears to be an argument (Fig. 2D) implying trimeric organization of LhcbM2 based on the designation “LhcbM,” but such reasoning does not make sense.

Reply: We sincerely appreciate your review and insightful comments. The LhcEs are species-specific identified in *E. gracilis*. Phylogenetic analysis revealed that LhcEs in *E. gracilis* PSI–LHC are more closely related to green algal Lhcb rather than to green algal Lhca (Supplementary Figure 8), indicating that *E. gracilis* LhcE may derive from green algal Lhcb. While LhcEs exhibit considerable structural similarities with green algal LhcbM, they display some variations in the loop structures (Supplementary Figure 10b). However, *E. gracilis* LhcbM2 and LhcbM8 share nearly identical structures with green algal LhcbM (Supplementary Figure 10a). In terms of structure, *E. gracilis* LhcbM2 and LhcbM8 are more similar to green algal LhcbM than to LhcE. This may provide some supports for the relation between *E. gracilis* LhcbM2/M8 and green algal LhcbM. *E. gracilis* LhcbM2/M8 form basal branches in the Lhcb phylogenetic tree (Supplementary Figure 8), in agreement with the phylogenetic analysis in the recent study (ref. 37), suggesting that *E. gracilis* LhcbM2/M8 evolved independently of green algal LhcbMs. *E. gracilis* LhcbM2/M8 and green algal LhcbMs may have common ancestor, whereas independent evolution weakens the relationship between them. As point out by the reviewer, the description that LhcbM2/LhcbM8 are closely related to the green algal LhcbM is insufficiently accurate. We

have checked the entire manuscript to avoid such inaccurate expressions.

Fig. 2D shows the oligomeric form of *E. gracilis* LhcbM2 (LHC-2) and LhcbM8 (LHC-3), which is identical to the assembly pattern of LhcbM1/M2/M3 that observed in the *C. reinhardtii* LhcbM trimer. As *E. gracilis* LhcbM2 and LhcbM8 share nearly identical structures with green algal LhcbM, *E. gracilis* LhcbM2 and LhcbM8 have the structural basis to form the oligomeric form similar to *C. reinhardtii* LhcbM trimer.

Reviewer 4:

This manuscript reports the determination of the structure of a PSI–LHCI–LHCII supercomplex of *Euglena gracilis* at 2.72 Å resolution using cryo-electron microscopy and presents the calculation of excitation energy transfer pathways within the supercomplex. The data presented in the present study are sound and the descriptions in the manuscript are clear. The supercomplex contained a “minimal” PSI core consisting of eight subunits and 15 LHC complexes. The minimal PSI core and the nomenclature of 15 LHCs (LHCBMs and LHCEs) were reported by Miranda-Astudillo et al. (Experimental Biology, 2025), which the authors cite as ref. 37. Recently, the structure of the PSI-LHC supercomplex of *Euglena gracilis* has been determined at a resolution of 2.82 Å by Kato et al (2025), as cited as ref 38 in this manuscript. This supercomplex contains the minimal PSI core and 13 LHC2. The structure is very similar but lacks two LHCs (LHC14/15). The structure determined in the present study is basically similar to that reported in ref 38. Thus, the most important finding on the structure of the supercomplex in the present manuscript is that two additional LHCs (LHCE5/8) were determined in the supercomplex structure. The significance of the present study is somewhat limited.

Reply: Thank you for your constructive comments and careful evaluation of our manuscript. We greatly appreciate your recognition of the soundness of our data and clarity of descriptions. We fully agree with your observation regarding the similarities between our structure and the recently reported *Euglena gracilis* PSI-LHC supercomplex (Kato et al., 2025, ref. 38). As you noted, our study identifies two additional LHC subunits in the supercomplex, which were not resolved in the 2.82 Å structure. Beyond this structural extension, we would like to emphasize several unique advances that enhance the significance of our work: First, we provide comprehensive analysis of excitation energy transfer (EET) pathways and pigment network dynamics. Second, the families of LHC proteins were assigned in our *E. gracilis* PSI–LHC structure which were not assigned in recently reported *E. gracilis* PSI–LHC. Third, the identification of LHCE5/8 and their specific interactions with the PSI core and other LHCs reveals a more complete antenna organization, shedding light on the adaptive evolution of light-harvesting systems in secondary endosymbionts.

We acknowledge that the overall structural framework shares similarities with ref. 38, but the higher resolution, functional EET analysis, and characterization of additional LHC subunits collectively deepen the understanding of *Euglena*’s unique photosynthetic machinery.

Specific points

1. L121 minimal PSI configuration: the authors use “minimal” and “simplified” to describe the PSI core consisting of eight subunits. It is better to use only one of them to prevent confusion for readers.

Reply: We have replaced “minimal” with “simplified”.

2. L149 “LhcbMX1-5” should be corrected as LhcbMX1-3 and LhcbMX5

Reply: We have revised this in the manuscript.

3. L239 “The PsaJ and PsaF subunits, which occupy the position of LhcbM observed in the green algal LHCII trimer” is correct?

Reply: We have revised the description as “The PsaJ and PsaF subunits associate with the LhcbM dimer (LHC-2/3), forming an organization similar to that of green algal LHCII trimer”